TOPICAL REVIEW

# Chemoreflex function in pulmonary diseases – A review

Kajal Kamra[1,2] (ID), Zhiqiu Xia[2], Irving H. Zucker[1] (ID), Harold Schultz[1] (ID) and Han-Jun Wang[1,2] (ID)

[1]*Department of Cellular and Integrative Physiology, University of Nebraska Medical Center, Omaha, Nebraska, USA*
[2]*Department of Anesthesiology, University of Nebraska Medical Center, Omaha, Nebraska, USA*

Handling Editors: Laura Bennet & Ken O'Halloran

The peer review history is available in the Supporting Information section of this article (https://doi.org/10.1113/JP286655#support-information-section).

**Abstract figure legend** Chemoreflex activation in respiratory diseases and COVID-19 populations.

**Abstract** The chemoreflex is a vital protective reflex that is crucial in restoring normal blood gas and pH levels. The carotid bodies are peripheral chemoreceptors activated by hypoxia (primarily), hypercapnia, acidaemia, temperature, lactate and potassium fluctuations. They are crucial for maintaining physiological conditions based on underlying causes and blood gas abnormalities. Central chemoreceptors in the brain stem are responsive to hypercapnia and contribute to blood gas regulation. Chemoreceptors regained attention during the SARS-CoV-2 (COVID-19) pandemic, potentially contributing to disease severity. In conditions like chronic intermittent hypoxia, chemoreflex sensitization may lead to respiratory disorders, impacting quality of life.

Kajal Kamra is a postdoctoral research associate at the University of Nebraska Medical Center, specializing in pulmonary physiology and pharmaceutical sciences. Her research spans carotid body chemoreflexes, acute lung injury and the development of inhalable gene therapies for chronic lung diseases. She designs advanced nanocarrier systems for targeted, non-viral delivery of gene-modulating tools to the lungs. Dr Kamra's work aims to advance regenerative nanomedicine and improve therapeutic strategies for respiratory disorders. She holds a PhD in Integrative Physiology and Molecular Medicine and is dedicated to bridging basic science with translational innovation.

The Journal of Physiology

There has been substantial research/evidence linking chemoreflex dysfunction to cardiovascular illnesses such as heart failure and hypertension. However, little emphasis has been placed on the role of the chemoreflex in mediating cardiopulmonary dysfunction during and after pulmonary disorders such as acute lung injury. The current review summarizes recent advances in the study of chemoreflex malfunction and the underlying mechanisms in acute and chronic lung diseases such as COVID-19 infection, chronic obstructive pulmonary disease, asthma, obstructive sleep apnoea and acute lung injury.

(Received 27 November 2024; accepted after revision 9 July 2025; first published online 31 July 2025)
**Corresponding author** Han-Jun Wang: Department of Anesthesiology, University of Nebraska Medical Centre, Omaha, NE 68198, USA.    Email: hanjunwang@unmc.edu

## Introduction

The chemoreflex is a fundamental protective reflex that is activated to restore normal levels of blood gases or pH in the body. There are two types of major chemoreflexes – the peripheral and central. The peripheral chemoreceptors are considered to be located in the carotid bodies (also known as glomus caroticum) at the bifurcation of the common carotid artery and aortic bodies at the aortic arch (Rakoczy & Wyatt, 2018). In the 19th and 20th centuries, a 'ganglion' (Iturriaga & Alcayaga, 2004) or a 'gland' (Nurse, 2010)-like structure at the bifurcation of the common carotid artery was recognized. The physiological role was assigned to this organ by Heymans in 1930 (Heymans & Heymans, 1927; Heymans et al., 1932). Heymans won his Nobel Prize for the discovery of the role played by the sinus and aortic mechanisms in the regulation of respiration. Although the main stimulus for activation of the carotid body is low $PaO_2$ (hypoxia), they are viewed as multimodal organs for they can also be activated in response to high $PaCO_2$ (hypercapnia), a decrease in pH (acidaemia), temperature, lactate and physiological fluctuations in extracellular potassium (Kumar, 2009). When the blood deviates beyond the normal range or set point of blood gas/pH levels, carotid body chemoreceptors become the primary defenders that restore physiological conditions. Often, during a chronic pathology such as heart failure, diabetes or sleep apnoea caused by repeated stimuli as in the case of chronic intermittent hypoxia (IH), the peripheral chemoreflex can be sensitized (Ponikowski et al., 2001; Peng & Prabhakar, 2004; Molkov et al., 2011; del Rio et al., 2013; Schultz et al., 2013; Marcus et al., 2014). The other type of chemoreceptors involved in maintaining blood gas homeostasis are the central chemoreceptors, located in the brainstem, primarily within the medulla. These receptors have long been recognized for their sensitivity to elevated arterial $CO_2$ (hypercapnia) and associated pH changes (Nattie & Li, 2012). Emerging evidence also indicates that certain populations of brainstem neurons, including those in the retrotrapezoid nucleus, may exhibit direct

or modulatory sensitivity to hypoxia, expanding their known role beyond $CO_2$ detection (Funk & Gourine, 2018; Gourine & Funk, 2017). Evidence suggests central chemoreceptors are involved in cardiovascular conditions such as heart failure and preserved ejection fraction conditions (Toledo et al., 2017). Chemoreflexes (peripheral and/or central) relay afferent information to the centres of the brain that send out efferent signals to various physiological systems including the diaphragm and chest muscles, to regulate breathing and to the autonomic nervous system to influence cardiovascular, renal, endocrine and immune function (Brognara et al., 2021; Hausberg & Grassi, 2007; Proczka et al., 2021; Schultz & Sun, 2000). Therefore, sustained activation of the carotid sinus nerves, a branch of the glossopharyngeal nerve that innervates the carotid sinus and carotid body in the case of a sensitized peripheral chemoreflex, or central chemoreceptors, may contribute to both respiratory disturbances and autonomic dysfunction. This heightened afferent signalling has been implicated in driving sympathetic hyperactivity, a hallmark feature in various pathologies such as heart failure, diabetes and metabolic syndrome, which in turn contributes to impaired physiological regulation and reduced quality of life (Andrade et al., 2015; Conde et al., 2018; Iturriaga, 2018; Iturriaga et al., 2014, 2016; McBryde et al., 2013; Niewinski, 2017; Paton et al., 2013; Prabhakar et al., 2015; Schultz et al., 2013). While the role of chemoreflex function in cardiovascular diseases such as heart failure and hypertension has been extensively studied in recent decades (McBryde et al., 2013; Narkiewicz et al., 2016; Schultz et al., 2013, 2015), earlier foundational work – particularly from the 1960s to 1980s by Nadel, Widdicombe and others, also explored chemoreceptor involvement in pulmonary conditions such as asthma and chronic obstructive pulmonary disease (COPD) (Nadel & Widdicombe, 1962). However, in more recent years, comparatively fewer studies have revisited the mechanistic role of chemoreflex pathways in chronic and acute pulmonary disease progression, especially within the context of emerging molecular

tools and immunomodulatory perspectives. However, less attention has been given to the role of chemoreflex in mediating cardiopulmonary dysfunction during acute and chronic pulmonary diseases. The current review will summarize the recent advances in exploring chemoreflex malfunction as well as the underlying mechanisms in acute and chronic lung diseases including COVID-19 infection, COPD, asthma, obstructive sleep apnoea (OSA) and acute lung injury (ALI).

## Anatomy and physiology of chemoreflex during physiological conditions

The primary functional units of the carotid body are type I (glomus) cells, which are derived from the neural crest and serve as the main chemosensors. In rats, each carotid body contains approximately 11,500 glomus cells (Laidler & Kay, 1975). These cells release neurotransmitters in response to hypoxia, hypercapnia and acidosis to activate afferent fibres of the carotid sinus nerve. Surrounding these chemosensitive glomus cells are type II (sustentacular) cells, which are glia-like cells thought to play a supportive role. However, emerging evidence suggests that type II cells are also involved in carotid body plasticity during sustained, (Gao et al., 2017) as well as chronic, intermittent hypoxic conditions (Caballero-Eraso et al., 2023).

In a series of studies, López-Barneo and colleagues demonstrated that type II cells can function as progenitors under chronic hypoxic conditions, differentiating into glomus-like cells or contributing to angiogenesis and structural remodelling of the organ (Gao et al., 2017; Ortega-Sáenz et al., 2013; Iturriaga et al., 2021). While the precise ratio of type I to type II cells may vary, some studies report a 4:1 ratio, favouring glomus cells (McDonald, 1981). The pivotal chemosensory role of type I cells was demonstrated by Verna et al. (1975), who showed that cryoablation of glomus cells abolished carotid body sensory activity.

To counteract changes in blood gases or pH, type I carotid body cells become activated through inhibition of potassium channels, leading to membrane depolarization. This depolarization opens L-type voltage-gated calcium channels, allowing calcium influx, which triggers the release of neurotransmitters. These neurotransmitters then activate afferent fibres of the carotid sinus nerve, forming a chemical synapse between type I cells and the sensory neurons (Fitzgerald, 2016; Gonzalez et al., 1994; Iturriaga et al., 2009; Ortega-Sáenz & López-Barneo, 2020; Prabhakar et al., 2015; Rakoczy & Wyatt, 2018). The chemoreceptor information is sent to the respiratory centre of the brain stem to cause specific reflexes to restore normal blood gas levels and pH in the body (Fitzgerald, 2016; Gonzalez et al., 1994; Iturriaga et al.,

2009; Ortega-Sáenz & López-Barneo, 2020; Prabhakar et al., 2015; Rakoczy & Wyatt, 2018). Individual clusters of glomus cells have been shown to have sensitivities and response profiles to hypoxia or low pH (Lu et al., 2013; Spiller et al., 2025; Zera et al., 2019). Some clusters only activate in response to either hypoxia (19% of glomus cells) or low pH (13% of glomus cells), while others may respond to both, but with varying intensities (Lu et al., 2013; Zera et al., 2019). Lu describes this heterogeneity as being reflected in differences in membrane potential, intracellular calcium levels and neurotransmitter release among glomus cells (Lu et al., 2013; Zera et al., 2019). Whether this occurs in response to all carotid body stimuli and in all mammalian species remains to be determined. The main types of neurotransmitters contained in carotid body type I cells are acetylcholine (ACh), dopamine, ATP and serotonin (Lu et al., 2013; Zera et al., 2019; Iturriaga et al., 2021; Zera et al., 2019).

Animal studies have provided ample evidence that relates specific neurotransmitters to their specific carotid body chemoreflex-mediated response. Various neurotransmitters can activate specific chemoreflex circuits. Fitzgerald and Zhang showed that ACh and ATP are likely to be accountable for the activation of sensory nerve fibres (Fitzgerald & Lahiri, 2011; Zhang et al., 2000). The glutamate and ATP neurotransmitters are involved in cardiovascular and sympatho-excitatory responses, while the cholinergic neurotransmitters mediate the respiratory component of the peripheral chemoreflex (Li et al., 2016; Zera et al., 2019). The afferent fibres of the carotid sinus nerve and the glomus cells can form chemical synapses of two types – bouton-like or/and calyx-like (Kondo, 1976; Nishi & Stensaas, 1974; Torrealba & Alcayaga, 1986).

Type I cells in rats express tyrosine hydroxylase (TH) and dopamine D2 receptors, consistent with their catecholaminergic phenotype. In humans, TH expression is present but relatively less robust (Ortega-Sáenz et al., 2013). The presence of nicotinic acetylcholine receptors and dopamine $\beta$-hydroxylase (DBH) – an enzyme for noradrenaline synthesis – has been more variably reported and is not a reliable distinguishing feature between cell types in all species (McDonald, 1981). Additionally, both type I and II cells may form gap junctions and engage in chemical signalling with each other and with afferent nerve endings, suggesting a more integrated model of carotid body communication (Ortega-Sáenz et al., 2013). Recent studies have proposed the existence of a tripartite synapse in the carotid body, analogous to that described in the central nervous system. In this model, type I (glomus) cells, afferent nerve terminals, and type II (sustentacular) cells form a dynamic signalling triad. Leonard et al. (2018) demonstrated that ATP released by glomus cells during hypoxia not only activates afferent fibres but also stimulates purinergic P2Y2 receptors on secondary ATP release from adjacent type II cells. This,

in turn, leads to intracellular calcium elevations and amplifies the chemosensory response. Type II cells also express pannexin-1 channels, enabling them to function as active gliotransmitters. This emerging model suggests that sustentacular cells do not merely play a passive supportive role but are instead functionally integrated into the chemosensory network, helping modulate the timing, intensity and plasticity of the carotid body's output (Leonard et al., 2018; Leonard & Nurse, 2022).

Activation of the peripheral chemoreflex by stimulation of carotid bodies causes a wide range of multi-system responses (Fig. 1). Peripheral chemoreflex activation is characterized by hyperventilation, an increase in phrenic nerve activity, and the co-activation of post-inspiratory and expiratory activities, brady-/tachycardia and sympatho-excitation. Evidence suggests that neurons from the carotid body primarily project to the nucleus tractus solitarius (NTS), particularly to its caudal medial subnuclei. However, retrograde tracing studies in rats have also revealed extra-NTS afferent projections to regions such as the area postrema, dorsal motor nucleus of the vagus, nucleus ambiguus and the caudal ventrolateral medulla (in the region of the nucleus retroambigualis) (Finley & Katz, 1992; Zera et al., 2019). These projections further support the existence of multiple reflex arcs sub-

serving distinct physiological responses. In a review by Zera et al. (2019), an extensive set of evidence supports the contention that there is a complex organization of carotid body connectivity to relevant brain stem reflex circuits to enable sensory information from the carotid body to be conveyed by a subset of afferents to a subset of neurons in the NTS that connect to distinct central reflex circuits, resulting in a wide diversity of the carotid body reflex evoked response. It is of importance to understand that some reflex pathways arise from the carotid body and can independently affect inspiratory, expiratory and sympathetic-mediated responses by targeting specific neural circuits, suggesting that there are at least two separate circuits that mediate the respiratory and sympathetic chemoreflex in the brain stem (Koshiya & Guyenet, 1994).

The brainstem encompasses a network of respiratory centres that interact to establish a coordinated pattern of breathing. Within this system, there are several key components (Fig. 1). The medullary respiratory centres, situated in the medulla oblongata, comprise: (a) the dorsal respiratory group, which initiates inhalation during regular resting breathing; and (b) the ventral respiratory group, responsible for both inhalation and exhalation. In the pons, there are the pontine respiratory centres,

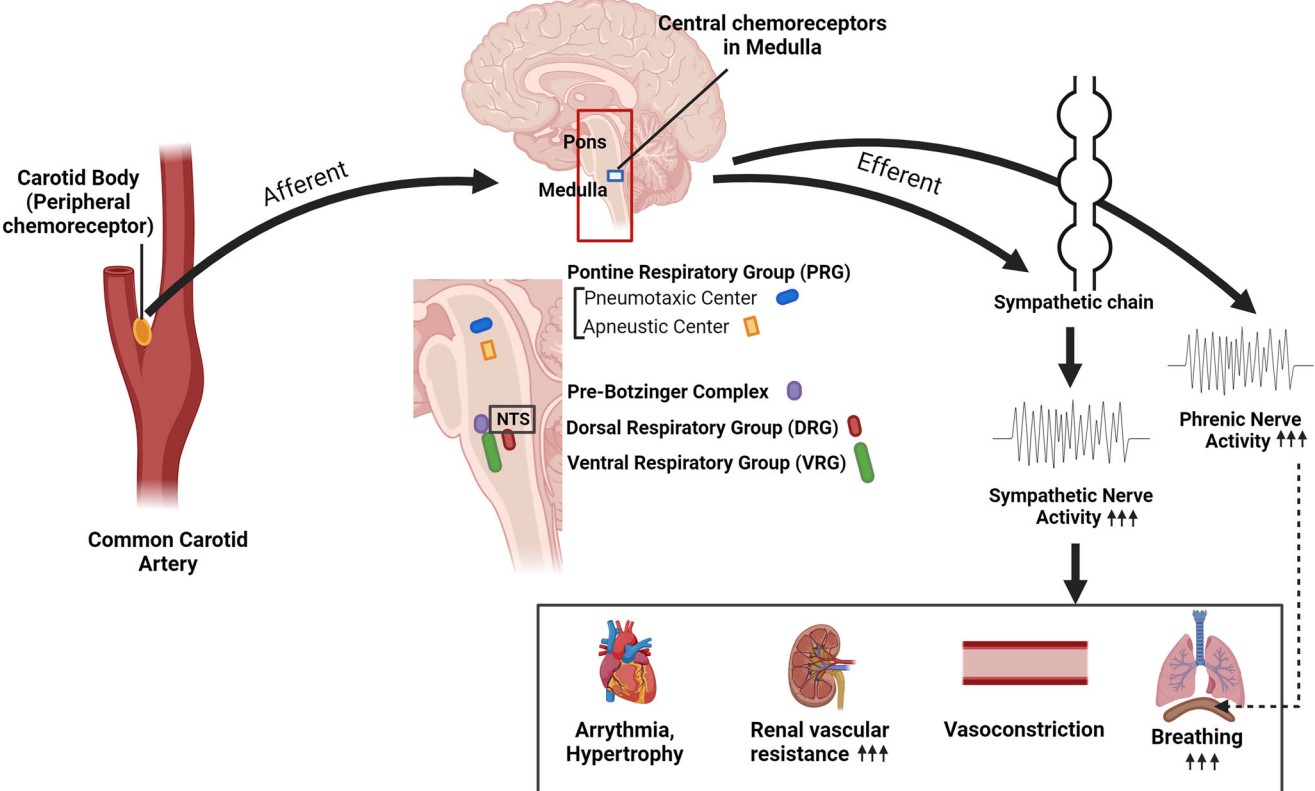

**Figure 1. Chemoreflex Arc and Multisystem Effects of Carotid Body Activation**
Illustration of a chemoreflex arc focusing on carotid body connectivity with the respiratory centre in the brain stem and efferent effects of chemoreceptor activation on a multiorgan system.

which consist of: (a) the pneumotaxic centre, tasked with fine-tuning the rate and depth of breathing; and (b) another pontine centre called the Apneustic centre, which plays a role in regulating breathing. It controls the intensity of breathing, giving positive impulses to the neurons involved with inhalation. The ventral respiratory column, located within the pre-Bötzinger complex, is made up of pre-inspiratory neurons. These neurons activate downstream premotor and motor neurons responsible for controlling the diaphragm, inspiratory muscles and muscles that regulate airway resistance.

To maintain precise control over breathing, these respiratory centres receive sensory input from various receptors, including the carotid bodies and central chemoreceptors. This sensory feedback enables the adjustment of breathing rates and depths in response to alterations in blood gas levels and pH. This intricate regulatory system ensures that homeostasis of efficient gas exchange and an adequate oxygen supply to meet metabolic demands is maintained.

## Carotid body sensitivity to inflammatory mediators

The carotid body is increasingly recognized as an immune-sensitive organ capable of detecting a wide range of circulating inflammatory signals. Studies have shown that it expresses receptors for several key cytokines, including IL-1$\beta$, IL-6, TNF-$\alpha$ and IL-10, in both rodents and humans (Iturriaga, 2023; Iturriaga et al., 2021; Sacramento et al., 2020). Under systemic inflammatory conditions, these cytokines increase carotid body afferent discharge and catecholamine release, contributing to sympathetic activation and respiratory instability. Pro-inflammatory cytokines such as IL-1$\beta$, IL-6 and TNF-$\alpha$ have been shown to inhibit oxygen-sensitive $K^+$ channels in glomus cells, resulting in membrane depolarization, elevated intracellular calcium levels and increased neurotransmitter release, thereby amplifying carotid sinus nerve activity and sympathetic output. For instance, IL-1$\beta$ suppresses $K^+$ currents and raises intracellular calcium in glomus cells, leading to heightened afferent firing from the carotid body (Shu et al., 2007). Similarly, IL-6 augments calcium signalling and catecholamine secretion in the carotid body, suggesting its excitatory role in chemosensory modulation (Fan et al., 2009). TNF-$\alpha$ can also amplify carotid body activity via TNFR1 receptors on glomus cells, feeding into a brainstem circuit that enhances sympathetic tone and contributes to systemic immune modulation – an effect that is diminished if the carotid body is removed (Katayama et al., 2022). These cytokines are upregulated in the carotid body during both chronic sustained and intermittent hypoxic exposure, further enhancing chemosensory responsiveness (Lam et al., 2008; del Rio et al., 2012). Notably, activation of toll-like receptors in the carotid body enhances inflammasome expression and IL-1$\beta$ release, which further sensitizes the chemoreflex arc (Ackland et al., 2013b).

In COVID-19, local inflammation in the carotid body – either from viral infiltration or cytokine storm – may further impair chemoreception (Lambermont et al., 2021; Villadiego et al., 2021). Moreover, SARS-CoV-2 has been detected in glomus cells of infected patients, with a high expression of Angiotensin-Converting Enzyme 2 (ACE2) (Lambermont et al., 2021; Villadiego et al., 2021). Even without direct viral invasion, the cytokine storm characteristic of severe COVID-19 could disrupt chemoreflex function (Aleksandrova & Danilova, 2010). Additionally, SARS-CoV-2-induced ACE2 depletion may tilt the local renin–angiotensin system toward Ang II/AT1R signalling, further enhancing chemosensitivity and sympathetic drive (El-Medany et al., 2024). This concept is underscored by studies showing that Ang I metabolites exert effects within the carotid body, and downregulation of ACE2 in the carotid body shifts the influence of Ang II to enhance carotid body chemosensitivity in heart failure (Patel & Schultz, 2013; Schultz, 2011).

More recently, researchers have shown that the carotid body also responds to inflammatory lipids, particularly lysophosphatidic acid (LPA). In models of allergic asthma, LPA, released in response to allergens or airway irritants, activates glomus cells via TRPV1 channels and LPA-specific receptors, increasing carotid sinus nerve activity and triggering vagally mediated bronchoconstriction (Jendzjowsky et al., 2018; Jendzjowsky, Roy, Iftinca et al., 2021). Blocking this pathway or removing the carotid body reduced airway hyperresponsiveness, pointing to its role in immune-related reflex control of the lungs. Together, these findings underscore the carotid body's emerging role as a neuroimmune sensor that links inflammation to breathing and autonomic function changes.

## Chemoreflex in SARS-CoV-2 (COVID-19)

In light of the recent COVID-19 pandemic caused by the novel SARS-CoV-2 virus, the carotid body chemoreceptors and their reflex have gained attention due to their potential involvement in regulating the severity of the disease that is primarily characterized by low $PaO_2$ (Chen et al., 2020; Sartini et al., 2020), in addition to symptoms including fever, dry cough, fatigue, ageusia, anosmia, headache and/or cerebral oedema (Guzik et al., 2020). Archer et al. suggest that COVID-19 may impair the body's oxygen-sensing mechanisms, particularly in the carotid bodies (Archer et al., 2020). This is supported by a case study where SARS-CoV-2 was found in the glomus cells of a COVID-19 patient's carotid body (Lambermont

et al., 2021), and the high expression of ACE2 in these cells (Villadiego et al., 2021). However, as these studies are limited to histological and molecular analyses, they do not offer direct evidence of carotid chemoreflex dysfunction. Without measured hypoxic or hypercapnic ventilatory responses in COVID-19 patients, compelling physiological evidence of chemoreflex involvement remains insufficient.

As described by Machado et al., the plasticity in peripheral arterial chemoreception function can occur at multiple levels including its different cell types, blood supply, afferent neurons, the central synapses with neurons in the NTS, central nervous reflex arc, and motoneurons in the brainstem and spinal cord (Machado & Paton, 2021). SARS-CoV-2 may impair carotid body function either directly, as indicated by Lambermont et al. (2021), or through the petrosal ganglia, potentially leading to taste loss, a symptom tied to petrosal neuron dysfunction. Additionally, neural damage in the brainstem areas controlling respiratory functions, similar to what was observed with SARS-CoV (Netland et al., 2008), is also possible. SARS-CoV-2 has been shown to enter the brain via the olfactory bulb and infect neurons, including those in the medulla oblongata that control autonomic functions. Reports suggest the virus can invade the brainstem in COVID-19 patients (Li et al., 2020). This may happen through peripheral vagal afferents and trans-synaptic transfer, consistent with findings from SARS-CoV and other coronaviruses. The virus's presence in blood, plasma, serum and organs suggests it can spread through the bloodstream (El-Medany et al., 2024). This spread is facilitated by the virus's presence in blood and organs, and may also involve the carotid body, which is highly vascularized and expresses ACE2. This invasion could disrupt peripheral chemoreception and potentially affect the solitary tract nucleus, which also expresses ACE2 and is susceptible to other coronaviruses (Soliz et al., 2020).

In addition to direct viral effects, recent findings discussed in the previous subsection highlight how the carotid body may contribute to the systemic pathophysiology of COVID-19 through its integration with immune and autonomic networks. In patients with mild COVID-19 viral infection who recovered well without hospitalization, the carotid body reflex is suggested to have overcome the hypoxic challenge inflicted by the disease (Machado & Paton, 2021). In other patients that required hospitalization or intensive critical care, it is suggested that the carotid body may have become overactive either before (such as in people with a pre-existing disease (Guzik et al., 2020), for example, obesity, diabetes, heart failure, hypertension, etc.) or as a result of the COVID-19 viral infection which led to intense autonomic and respiratory responses and subsequent pathophysiology affecting other organs

such as the heart, lungs and kidneys (Simonson et al., 2021). Abnormal excitability of carotid bodies (carotid body hyperreflexia) caused by related and non-related SARS-CoV-2 pathogen-associated molecular patterns and damage-associated molecular patterns impact both susceptibility to infection and the severity of disease (Ackland et al., 2013a; Holmes, 2020; Mkrtchian et al., 2020). A proportion of COVID-19 patients were also suggested to have desensitized or non-functional carotid bodies before (for example, people who underwent carotid body resection as a therapy to treat a previous disease) or during the viral infection (Villadiego et al., 2021). Such patients were seen to have 'silent hypoxia' and were believed to be at a much higher risk because the infection gets well-established before they realize they need to seek medical attention. Some respiratory viruses, including SARS-CoV-2, may suppress hypoxia-inducible factors, crucial for oxygen-sensing in the carotid body (Marchetti, 2020). This could explain the insufficient hyperventilatory response and mild hypocapnia in COVID-19 patients with silent hypoxia (Dhont et al., 2020; Gonzalez-Duarte and Norcliffe-Kaufmann, 2020; Tobin et al., 2020). Villadiego et al. (2021) propose that SARS-CoV-2 infection of the carotid body via ACE2 might lead to inflammation and death of glomus cells, diminishing the body's chemosensitivity.

## Obstructive sleep apnoea and chronic intermittent hypoxia

An increase in sympathetic tone is also seen in animal models and patients with OSA. Apnoea is the cessation of breathing due to a complete or partial pharyngeal collapse that results in a complete absence of airflow (apnoea) or a decrease in airflow (hypopnoea). These can lead to oxygen desaturation and an increase in arterial carbon dioxide (hypercapnia), an increase in intrathoracic pressure and sleep fragmentation (Sunderram & Androulakis, 2012). OSA is a sleep-breathing disorder that affects 10–20% of the adult population worldwide, 10–17% of whom are male and 3–9% are female, aged 30–70 years (Peppard et al., 2013). The severity of OSA is defined by the number of apnoeic and hypopnoeic episodes per hour (Apnoea-Hypopnea Index, AHI) of sleep. An AHI of $<5$/h of sleep is considered normal and can be found in any healthy subject. An AHI of 5–15 is mild OSA, 15–30 is moderate, and $>30$ is considered severe OSA (Somers et al., 2008). IH is a hallmark feature of sleep apnoea. This change in blood gas levels, depending on the severity of OSA, triggers the activation of chemoreflexes. Interestingly, with comorbidity (such as pre-existing hypertension), which is common and is associated with people exposed to IH, sympathetic nerve activity (SNA) is 6–12-fold higher than in normotensive patients with

OSA (Somers et al., 1988). This comparable, significant increase is believed to be explained, in part, by impairment of baroreflex function in hypertensive patients. Although OSA is highly prevalent in obese individuals, studies have shown that peripheral chemoreflex sensitization in normotensive OSA patients cannot be solely attributed to obesity (Mansukhani et al., 2014; Narkiewicz, van de Borne, Cooley et al., 1998; Smith et al., 1996). Obese OSA patients demonstrate disproportionately heightened central chemoreflex sensitivity, whereas lean OSA patients exhibit more pronounced peripheral chemoreflex responses (Mansukhani et al., 2014). This distinction suggests that mechanisms underlying chemoreflex sensitization may differ by body composition and that central chemoreflex pathways may play a dominant role in ventilatory control among obese OSA individuals.

Several sleep studies have found that the activation of the chemoreflex, sympathetic activity and alterations in cardiovascular parameters during sleep in patients with OSA do not show a strong association with the REM and NREM sleep stages. This varies from patient to patient depending on the severity of OSA. During wakefulness, normoxic OSA patients exhibit a tonic activation of the chemoreflex, which is shown to be inhibited with the administration of 100% oxygen (Kara et al., 2003; Narkiewicz, van de Borne, Montano et al., 1998; Xie et al., 2000). The chemoreflex activation by hypoxia or hypercapnia led to the initiation of hyperventilation and consequent stretch of thoracic afferents that result in inhibition of chemoreflex-mediated sympathetic activation, micro-arousals and restoration of airflow (Kara et al., 2003).

As mentioned above, a key pathological feature of OSA is the resulting IH, which plays a central role in mediating the cardiovascular and neuroinflammatory sequelae associated with the disorder. IH occurs in conditions such as OSA, where repeated airway obstructions during sleep cause cycles of low oxygen and reoxygenation, as well as in individuals who frequently travel between low and high altitudes, such as miners, military personnel and athletes. IH causes hypertension, increased hypoxic ventilatory response, autonomic imbalance, decreased baroreflex gain and increased arrhythmias and atrial fibrosis (del Rio et al., 2016). Increased peripheral chemoreflex sensitization is one of the main contributing factors to sympatho-excitation. Intermittent hypoxia (acute or chronic) has been shown to increase SNA in animal models (Bao et al., 1997; Zoccal et al., 2007, 2008) and in humans (Smith & Pacchia, 2007; Tamisier et al., 2011). A study by del Rio et al. showed that a bilateral carotid body ablation in rats reduced their blood pressure, reduced ventilatory response to hypoxia, and restored cardiac autonomic and baroreflex functions (del Rio et al., 2016). They provide evidence that the carotid body plays a crucial role in IH-induced hypertension as ablation of carotid bodies not only reduced blood pressure and normalized the hyperventilatory response to hypoxia but also restored cardiac autonomic and baroreflex functions without affecting systemic oxidative stress (del Rio et al., 2016). Clinically, intermittent hypoxia is more potent in augmenting SNA than sustained hypoxia (Smith & Pacchia, 2007). IH induces sensory long-term facilitation (sLTF) of the carotid body, i.e. a long-lasting activation of baseline carotid body activity (Grebe et al., 2006; Peng et al., 2003; Sunderram & Androulakis, 2012). Other than its action on peripheral chemoreceptors, it has also been shown to induce alterations in the central sites of sympathetic regulation (Neubauer & Sunderram, 2004; Sunderram & Androulakis, 2012). In addition to changes in chemoreflex sensitivity, other effects of an enhanced SNA include changes in cardiovascular responses. Multiple studies done in dogs (Brooks et al., 1997) and rats (Fletcher, Lesske, Qian et al., 1992) provide evidence of persistent hypertension due to elevated sympathetic tone (Fletcher, Lesske, Culman et al., 1992) caused by IH. Hypertension in these animals was shown to be prevented by renal sympathectomy or adrenal medullectomy (Bao et al., 1997). There is evidence of IH-induced peripheral hypoxic chemosensitivity in various species. In rats (Prabhakar et al., 2007) exposed to intermittent hypoxia for 5 min, 8 h per day for 10 days, there was an increase in carotid sinus nerve activity, increased sympathetic vasoconstrictor outflow, and enhanced chemoreflex-induced sympatho-excitation (Fig. 2). Similarly, mice exposed to 10 days of intermittent hypoxia for 8 h per day also exhibited augmented carotid body responses to intermittent hypoxia. Eight hours of intermittent hypoxia in cats for 4 days also increased the baseline chemosensory discharge and carotid body responses (Rey et al., 2006).

Kara et al. found that this inhibition was more pronounced during activation of peripheral chemoreceptors than central chemoreceptors (Kara et al., 2003). Studies by Fletcher et al., Peng et al., and del Rio et al. provide strong evidence of the involvement of sensitized carotid body chemoreflex in IH-exposed hypertension rodent models, which are characteristic of OSA (Fletcher, Lesske, Behm et al., 1992) (Peng et al., 2003; del Rio et al., 2016). In one such study, bilateral denervation of the carotid sinus nerve before exposure to IH was shown to abolish enhanced blood pressure responses in rats (Fletcher, Lesske, Behm et al., 1992). In another study, the elimination of carotid body chemosensory input to the brainstem in IH-exposed rats immediately normalized elevated blood pressure, restored baroreflex sensitivity and normalized heart rate variability (del Rio et al., 2016). The most used therapy to manage OSA is continuous positive airway pressure (CPAP), which has resulted in decreased sympathetic activity and attenuated blood pressure surges in sleep (Narkiewicz, van de Borne, Cooley et al.,

1998; Querido et al., 2010). It has also been shown to significantly attenuate muscle sympathetic nerve activity after 6 months and 1 year of CPAP treatment in OSA patients compared with untreated patients (Imadojemu et al., 2007; Narkiewicz et al., 1999).

Researchers have found multiple mechanisms that can induce carotid body potentiation and autonomic alterations during IH and OSA. In both animal (Peng & Prabhakar, 2003) and human studies (Jordan et al., 2006; Lavie et al., 2004), the hypoxia–reoxygenation cycles have been shown to result in the elevation of reactive oxygen species (ROS) which is proposed as a significant contributing factor to potentiate chemosensitivity (Sunderram & Androulakis, 2012). The NOX family of enzymes plays an important role in transporting electrons across the plasma membrane to generate ROS. Following IH, there is evidence of both cytosolic and mitochondrial ROS (Prabhakar et al., 2010) generation in the glomus cell of the carotid body mediated specifically by activation of NOX2 which was observed to be reduced after treatment with inhibitors of NOX and superoxide dismutase suggesting the involvement of ROS (Prabhakar et al., 2010). ROS are a necessary requirement for an augmented carotid body chemoreflex during hypoxia post-IH (Pawar et al., 2009).

Another factor that can regulate carotid body potentiation is the hypoxia-inducible factor-1 (HIF-1) (Semenza, 2011), a transcription factor that regulates the expression of various genes under hypoxic conditions. There are two isoforms of HIF-1, namely, HIF-1 alpha and HIF-1 beta that dimerize under hypoxic conditions to activate the transcription of several target genes. HIF-1 alpha heterozygous mice deficient in the HIF-1 alpha subunit did not show carotid body sLTF, enhanced carotid body responses to hypoxia, and elevated ROS levels following IH exposure, suggesting its involvement in carotid body potentiation post-IH (Sunderram & Androulakis, 2012). Chronic intermittent hypoxia has also been shown to increase the levels of vasoconstrictor peptides such as endothelin-1 (ET-1) that are expressed in the glomus cells and blood vessels of the carotid body (Rey et al., 2007). An increase in the expression of both ET-1 and the $ET_A$ receptor has been documented in IH (Chen et al., 2002). Use of scavengers of free radicals prevents the augmented sensory responses and increase in ROS and ET-1 levels and ETS mRNA following IH exposure (Pawar et al., 2009). Nanduri et al. also found an increase in oxidative stress in the carotid body and the adrenal medulla due to DNA methylation-dependent suppression of genes that encode for antioxidant enzymes (Nanduri et al., 2019). Angiotensin-II (Ang II) has also been implicated to directly enhance peripheral chemosensitivity (Schultz, 2011). The link between enhanced SNA and the renin–angiotensin–aldosterone system is well established. Similar to ET-1, Ang II levels and the expression of Ang II type 1 (AT1) receptor in the carotid body post-IH exposure (Lam & Leung, 2002) are increased.

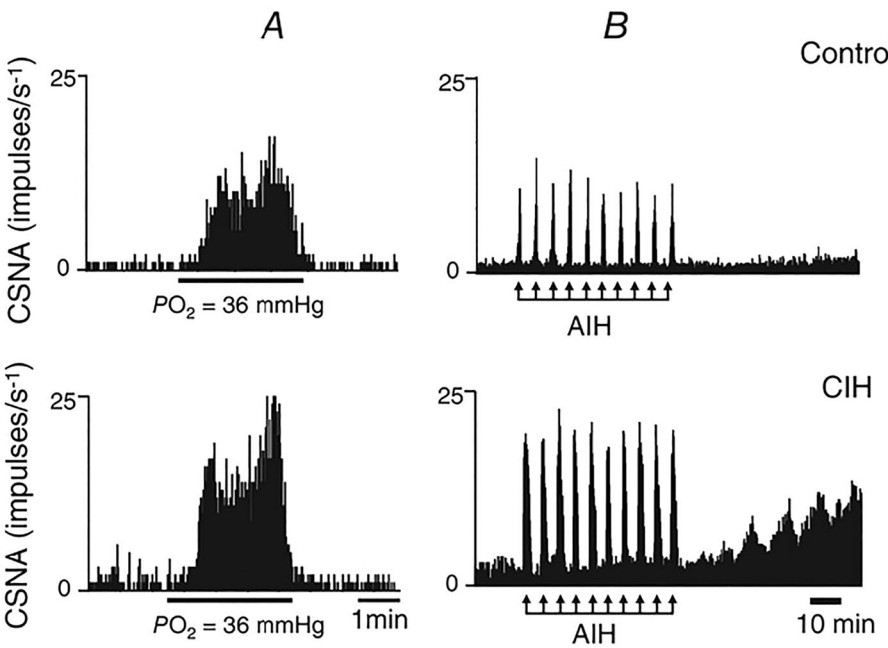

**Figure 2. Sensitization and Long-Term Facilitation of Carotid Body Response After Intermittent Hypoxia**
Examples of sensitization of carotid body response to hypoxia (*A*) and sensory long-term facilitation (*B*) in rats subjected to 10 days of intermittent hypoxia. Adapted from *Respir. Physiol. Neurobiol.*, *157*, 148–153, 2007 with permission.

Carotid body chemoreceptor activity is also modulated by endothelial nitric oxide synthase (eNOS) and neuronal NOS (nNOS) that are present in the carotid body (Iturriaga et al., 2000) (Schultz et al., 2012). Regardless of the specific mechanism, the endogenous NO production is low in patients with sleep apnoea (Haight & Djupesland, 2003; Weiss et al., 2012). NO is known to suppress peripheral chemoreflex activity by hyperpolarizing glomus cells (Li et al., 2010) and through inhibitory efferent signalling via the glossopharyngeal nerve, potentially modulating chemosensory output through changes in ion channel activity and neurotransmitter release. Recently, Bock et al. (2021), provided evidence that augmented circulating NO bioavailability via inorganic nitrate supplementation can attenuate a diurnal rise in systolic blood pressure in patients with mild to moderate OSA which may be mediated by a reduction in peripheral chemoreflex sensitivity as shown by a blunted ventilatory response to hypoxia.

### Chronic obstructive pulmonary disease

COPD is a lifelong, polygenic, systemic disease. Patients with COPD may have obstructive bronchiolitis, emphysema, or a combination of both (Lareau et al., 2019). COPD patients show limited lung function, a progressive and partially reversible airway obstruction, marked dyspnoea, systemic inflammation, cachexia, skeletal muscle dysfunction, cardiovascular and osteoskeletal changes and exercise intolerance (Agustí et al., 2020). Due to the heterogeneity in COPD patient phenotypes, disease traits and mechanisms behind COPD (Agustí & Hogg, 2019; Agusti et al., 2010; Cosio et al., 2009) Agusti suggested that COPD be considered as a 'syndrome' rather than a 'disease' (Celli & Agustí, 2018). One of the main consequences of COPD is the increase in resting sympathetic activity (van Gestel & Steier, 2010). An increase in SNA is predictive of the severity, mortality and hospitalizations in COPD (Andreas et al., 2014). The symptoms or traits of the disease change as the severity of the disease progresses. In the case of mild COPD, an increased sympathetic drive is reported that increases resting heart rate and depresses baroreflex sensitivity (Haider et al., 2009).

There is no consensus in the existing literature regarding the changes in chemoreflex drive in COPD (Hida, 1999). There are controversial findings. Some studies report that in the case of humans with mild COPD, chemosensitivity to hypoxia/hypercapnia can be either enhanced or blunted (Altose et al., 1977; Bradley et al., 1979; Hida, 1999; Kelsen et al., 1979). Studies provide evidence of an abnormal ventilatory drive to exercise in mild COPD patients. This may not be caused entirely due to limited lung function or mechanics suggesting that enhanced carotid body chemo-

reflex activity/sensitivity may be involved (Stickland et al., 2016). 100% $O_2$ administration to inhibit the peripheral chemoreflex in mild COPD patients was shown to reduce minute ventilation transiently, indicating greater tonic peripheral chemoreceptor activity in COPD (Stickland et al., 2016). Stickland et al. also found that the carotid body chemoreflex sensitivity was directly related to arterial stiffness (Bestall et al., 1999; Vedam et al., 2009) in COPD and was seen to reduce with administration of 100% $O_2$ (Stickland et al., 2016). However, long-term use of hyperoxia can be detrimental to patients as it can cause secondary effects including oxidative stress, central stimulation and vasoconstriction (Bartels et al., 2004; Dean et al., 2004; McNulty et al., 2007), making the situation even worse. In severe COPD, most studies report a blunted peripheral chemoreflex drive (Hida, 1999; Haider et al., 2009). Blunted chemoreceptor activity puts these patients at a much higher risk of experiencing fatal episode(s) (Hida, 1999; Haider et al., 2009). Treatment with bronchodilators, supplemental oxygen administration, lung volume reduction surgery and the use of beta-2 agonists improves the ventilatory drive in such patients (Bartels et al., 2004; Costes et al., 2004; Lareau et al., 2019). Low ventilatory drive to hypercapnia in severe COPD patients could lead to the development of hypercapnia, nocturnal hypoxaemia (Fleetham et al., 1980) and elevated pulmonary artery pressure (Boysen et al., 1979). The chemoreflex activity in COPD is still poorly understood and more studies are required to obtain further clarity on chemoreflex activity and the underlying mechanisms in COPD patients so a treatment regime can be modified for patients with this disorder.

### Asthma

Asthma is a disease of chronic inflammation in the airways caused by contraction of the airway smooth muscles and copious mucus secretion which leads to intermittent symptoms of wheezing, cough, chest tightness and dyspnoea with variable expiratory airway obstruction (Reddel et al.). Clinical diagnosis is done based on the symptoms presented and the expiratory airflow obstruction measured by pulmonary function tests (Price et al., 2015). The onset of asthma can occur at any age (Asher & Pearce, 2014; Alith et al., 2015; Price et al., 2015) and with an increase in age, it becomes difficult to differentiate asthma from COPD or asthma–COPD overlap syndrome (Freiler, 2015; Price et al., 2015). Asthma is more common in children (Enilari & Sinha, 2019) but affects 7–9% of the US population older than 65 years, causing a 4-times higher risk of mortality in the older population (Lee et al., 2012; Onufrak et al., 2008; Tattersall et al., 2018). Mortality can occur either when the attack lasts for an extended duration such as days or weeks,

or, during the occurrence of fulminant episodes (Arnold et al., 1982; Sears et al., 1985).

During an acute severe asthmatic attack, there is an increased work of breathing due to blatant functional convulsions with airflow obstruction and air trapping leading to ventilation–perfusion mismatch producing hypoxaemia (Town & Allan, 1989). The ventilatory drive plays a key role in determining the severity of asthma and may be influenced by several factors such as chemosensitivity, basal arterial oxygen or carbon dioxide tension, mechanical impedance and respiratory muscle dysfunction (Hida, 1999). In asthmatics, hypoxaemia alone is considered a mild stimulus to respiration but the presence of hypercapnia can enhance its effect on the brain's respiratory centre (Town & Allan, 1989). Airway obstruction, which is a characteristic of asthma, is known to increase the central drive, inducing an increase in the hypercapnic ventilatory response in patients with asthma (Hida, 1999). Pharmacologically induced bronchoconstriction has been demonstrated to affect central respiratory output, inspiratory motor activity and breathing patterns (Hida, 1999). As reported by Fujimore et al. (1996), ventilation first decreases and then increases in both healthy and asthmatic subjects. When respiratory resistance was doubled, the change in ventilation was doubled in asthmatic subjects when compared with healthy subjects. These investigators describe this difference in ventilatory response between healthy subjects and asthmatics to be caused by different increases in respiratory drive caused by irritant receptors, not chemical stimuli (Fujimori et al., 1996). However, more often than not, in such a situation, an increase in ventilation is observed (Town & Allan, 1989). This increase is believed to be caused by a combination of chemical and mechanical stimuli (Town & Allan, 1989). Respiratory muscles respond to the increased workload created by airflow limitation, and vagal receptors are likely stimulated by mediators released in the airways (Kelsen et al., 1979; Rebuck & Slutsky, 2011). A mechanistic understanding of this is not well understood but it seems possible that the neuromuscular output during hypercapnia increases due to an increased airflow obstruction and/or the vagal afferent activity may increase in response to hypercapnia (Hida, 1999).

The chemoreflex response in asthmatic subjects remains controversial and confusing. Some human studies suggest that asthmatics as a group demonstrate a normal response to hypoxia and hypercapnia (Cosgrove et al., 1976; Kelsen et al., 1979). However, in a study by Hutchison et al. (Hutchison & Olinsky, 1981), two young adults with severe asthma and frequent episodes of respiratory failure had a normal response to hypoxia but one had a low response to hypercapnia. Three young asthmatic children in the same study demonstrated a blunted response to hypoxia but a normal response to hypercapnia. More cases from other patient cohorts in a study by Kikuchi et al. (1994) indicate that asthmatics with fatal or near-fatal attacks experience a blunted/reduced hypoxic response as compared with asthmatics without such attacks or normal subjects. Their mean vital capacity and FEV1 measured in asthmatics were higher than in normal subjects and much higher in asthmatics with near-fatal attacks. In a separate study, no significant differences in the duration and severity of asthma and airway hyperreactivity were found in asthmatics with and without fatal attacks (Takishima et al., 1981). The hypercapnic response in those patients was normal. These data contradict the findings of Hutchison's study.

Hypoxic drive is regulated and is dependent on the carotid bodies. Blunted carotid body chemoreception may be caused by a dysfunctional or malfunctioning carotid body during asthma with near-fatal episodes (Barnes, 1994) or be attributed to a bilateral carotid body resection (Honda et al., 1976). Carotid body resection causes a decrease in the ventilatory drive to hypoxia and is therefore not advised in humans (Reznikov, 2021). A well-characterized cohort of asthmatic children demonstrated carotid arterial injury in a study by Tattersall et al. (2018). What causes the malfunctioning of carotid bodies in such patients, or the cause–effect relationship between carotid bodies and asthma is not clear. Response to hypoxia or hypoxic ventilatory drives in normal, healthy individuals varies (Barnes, 1994). Some healthy individuals have a reduced respiratory response due to an inheritance of inappropriately low, familial responses to hypoxia or hypercapnia (Town & Allan, 1989).

Asthmatic patients with near-fatal asthma also experience a reduced perception of dyspnoea. Dyspnoea is subjective and is difficult to assess (Lowell, 1973). Decreased chemosensitivity accompanied by reduced perception of dyspnoea is very common and is a fatal combination (Kikuchi et al., 1994). Delay in initiating appropriate response results in a dysfunctional defence mechanism against profound hypoxaemia and airway narrowing that results in a risk of respiratory arrest or sudden death (Enilari & Sinha, 2019). Although studies suggest that there is a correlation between the impairment in hypoxic drive and the perception of dyspnoea, the mechanism connecting the two is not well understood (Barnes, 1994). The perception of dyspnoea is affected by variable factors and involves several different central afferent pathways (Hida, 1999) and respiratory motor outputs (Chonan et al., 1990). Some believe that airflow obstruction may contribute to the reduced perception of dyspnoea due to the inability to increase ventilation (Barnes, 1994). Current treatment therapy for asthma includes the use of beta-2-agonist and corticosteroid therapy, both of which reduce the perception of dyspnoea (Hida, 1999).

Recent studies have further strengthened the evidence that the carotid body plays an active role in asthma pathophysiology, beyond its traditional role as a chemosensor. Jendzjowsky et al. (2018) demonstrated that activation of the carotid body by allergen exposure leads to vagally mediated bronchoconstriction, establishing a mechanistic link between carotid body activation and asthma-like symptoms in a murine model (Jendzjowsky et al., 2018). In a follow-up study, Jendzjowsky et al. (2021) provided compelling evidence that blood-borne inflammatory mediators, such as IL-4, IL-5 and IL-13, can directly sensitize carotid body glomus cells via IL-5 receptor expression, enhancing afferent signalling from the carotid body (Jendzjowsky, Roy, & Wilson, 2021). These findings suggest that the carotid body may serve as an immunosensor, contributing to airway hyper-responsiveness and altered ventilatory drive in asthma via neuroimmune signalling pathways. This could provide a mechanistic basis for the blunted hypoxic response and reduced perception of dyspnoea observed in patients with near-fatal asthma episodes.

### Acute lung injury

Acute lung injury (ALI) accounts for at least 10% of ICU admissions with high morbidity and mortality. It remains an economic burden and affects approximately 200,000 new cases each year in the US alone (Mowery et al., 2020). It causes alveolar damage, disruption of the normal endothelial barrier and invokes perturbations of ventilatory control (Young et al., 2019). The impairment in gas exchange caused due to ALI induces hypoxaemia, hypoxic ventilatory response (HVR) and an increase in resting respiratory rate (Bernard et al., 1994; Jacono et al., 2006; Spinelli et al., 2020). ALI may be caused directly, because of smoke inhalation, aspiration, etc., or indirectly due to sepsis, drug reaction, etc. (Shaver & Bastarache, 2014). The mechanism of abnormal breathing in ALI is not clearly understood. Evidence suggests that the carotid body's chemosensing function is altered during ALI (Huxtable et al., 2011; Jacono et al., 2006). Some groups have studied the HVR during the 'early' stage of ALI (Huxtable et al., 2011; Jacono et al., 2006). For example, Jacono et al. measured chemoreflex function in response to hypoxia at day 5 post-ALI and reported sensitization of HVR (Jacono et al., 2006). The chemoreflex function at an early stage of ALI seems to depend on the severity of the disease. In a rodent model, our group showed that in the case of a mild ALI induced by bleomycin, at week 1 post-ALI, the chemoreflex response to hypoxia was sensitized, while in the case of a severe ALI, it was blunted (Fig. 3) (Kamra et al., 2022). Similar results were reported by a different group of researchers who studied the chemoreflex at week 1 post-ALI in an Lypopolysaccharide (LPS)-rodent model (Huxtable et al., 2011). Kamra et al. also provide experimental evidence of chemoreflex sensitization in both mild and severe ALI in response to both hypoxia and hypercapnia during the 'recovery' stage of ALI when the resting respiratory rate was partially normalized (Kamra et al., 2022). They also show that the chemoreflex remains sensitized in both male and female bleomycin-induced ALI rats (Kamra et al., 2024).

A sensitized chemoreflex affects the overall quality of life due, in part, to excessive sympathetic outflow (Conde et al., 2014; Cunha-Guimaraes et al., 2020; del Rio et al., 2017) as explained in the introduction section of this paper. What causes the chronic sensitization of chemoreflex post-ALI is not well understood. Some speculate that neuro-inflammation by ALI-induced cytokines in the NTS may alter the ventilatory pattern after ALI (Getsy et al., 2019; Hsieh et al., 2020; Litvin et al., 2020). The postganglionic sympathetic axons of the superior cervical ganglion (SCG) innervate carotid bodies (Iturriaga Agüera et al., 2016). It has been shown that if stimulated electrically, the sympathetic efferent nerves that originate in the SCG can sensitize the chemoreflex in hypertensive and normotensive rats (Felippe & Paton, 2021; Felippe et al., 2022; Getsy et al., 2021). We speculated that the involvement of SCG is involved in alterations of carotid body chemoreflex function post-ALI. A bilateral superior cervical ganglionectomy (SCGx) in a bleomycin-induced ALI rodent model significantly attenuated the sensitized chemoreflex post-ALI (Fig. 4) (Kamra et al., 2023). As discussed by Kamra et al. (2023), a release of increased ATP and noradrenaline from SCG to the carotid body may activate purinergic/adrenergic receptors in the nerve terminals of chemosensory cells of the carotid body to cause chemoreceptor sensitization post-ALI. Another potential mechanism is oxidative stress-induced changes in carotid body function. Like chronic heart failure (Andrade et al., 2015), ALI also leads to systemic inflammation and hypoxia, which may lead to an increase in Ang II-dependent oxidative stress which could alter carotid body function in ALI (Imai et al., 2005). The above mechanisms need to be investigated in the future for a full understanding of the functions and changes of chemoreflex in ALI conditions.

### Carotid body denervation – a therapy to treat pulmonary diseases?

**Therapeutic promise in disease states.** Carotid body denervation (CBD) is being actively revisited as a potential therapeutic strategy in pulmonary and cardiovascular diseases characterized by chemoreflex hypersensitivity, such as asthma, COPD, sleep-disordered

breathing and chronic heart failure. In these conditions, hyperactive carotid bodies are believed to contribute to disordered breathing patterns, sympathetic overactivation and increased morbidity. In a rat model of chronic heart failure, del Rio et a. (2013) demonstrated that selective carotid body ablation reduced sympathetic outflow, stabilized ventilation, lowered arrhythmia risk and prolonged survival, highlighting its cardiopulmonary benefits (del Rio et al., 2013). Similarly, Marcus et al. (2014) reported that CBD enhanced baroreflex sensitivity, restored autonomic balance, and reduced disordered breathing in rats with congestive heart failure (Marcus et al., 2014). These results suggest that targeted CBD may offer symptom relief and disease modification in settings where overactive carotid body signalling is pathogenic.

**Physiological risks and historical evidence.** Despite its potential in specific disease contexts, CBD presents important physiological and clinical concerns, especially in otherwise healthy individuals. In animal studies,

short-term effects of CBD include hypoventilation, apnoea, a reduction in the HVR and diminished $CO_2$ sensitivity (Timmers et al., 2003). Mouradian et al. (2012) further demonstrated in rats that bilateral CBD causes chronic hypoventilation and attenuated ventilatory responses to both hypoxia and hypercapnia, underscoring the carotid bodies' critical role in respiratory homeostasis (Mouradian Jr et al., 2012).

Importantly, in many mammals, variable restoration of chemoreflex function after CBD has been reported, potentially due to compensatory inputs from aortic and abdominal chemoreceptors or through central plasticity (Breslav & Konza, 1975; Bisgard et al., 1976; Bisgard et al., 1980; Forster et al., 2000; Martin-Body et al., 1986; Pan et al., 1998; Serra et al., 2001; Timmers et al., 2003).

Historically, in humans, multiple carotid body resections have been attempted (Honda, Myojo et al., 1979; Wade et al., 1970). However, clinical acceptance remains polarized – while some clinicians report patient improvement, others observe little to no benefit (Winter, 1973). Other than the psychological benefit, unilateral

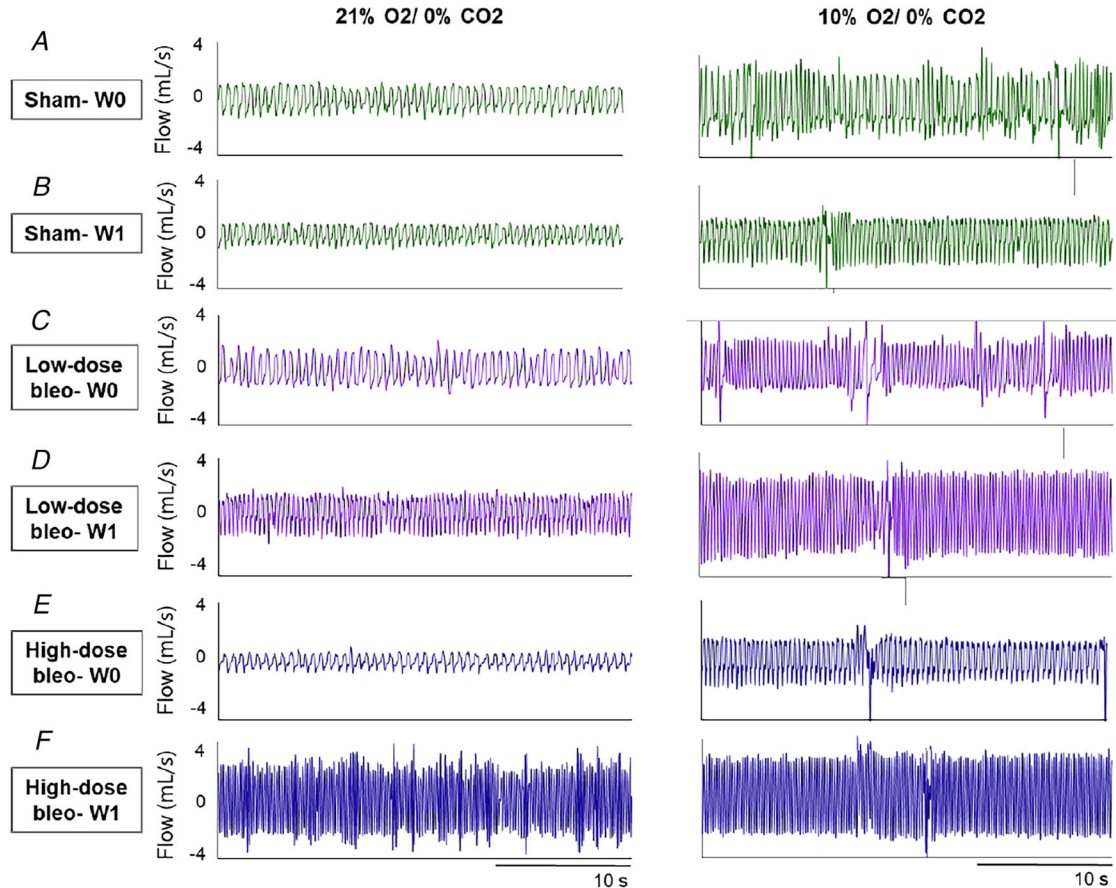

**Figure 3. Resting Breathing Responses to Hypoxia in Sham and Bleomycin-Treated Rats Over Time**
Representative recordings of resting breathing at normoxia (on left) and 10% hypoxia (on right) obtained in one rat per experimental group: (*A*) and (*B*) Sham (in green) at W0 and W1, respectively; (*C*) and (*D*) low-dose bleo (in purple) at W0 and W1, respectively; (*E*) and (*F*) high-dose bleo (in blue) at W0 and W1, respectively. Adapted from *Front Physiol. 2022 Oct 20;13:1 009 607* with permission.

removal is frequently said to result in no improvement (Winter, 1973). Bilateral resection of the carotid body has been usually attempted as a desperate measure when unilateral resections fail (Winter, 1973).

Importantly, surgical technique significantly influences outcomes. For example, a bilateral carotid body resection associated with a bilateral resection of the lateral sinus nerve plexus is not ideal, physiologically (Holton & Wood, 1965). Resections without damaging the lateral plexus have been reported as useful for the treatment of asthma and emphysema (Winter, 1972). As summarized by Timmers et al. (2003), the results from these procedures do not impair baroreflex function, though patients demonstrate limited ventilatory responses to both hypoxaemia and hypercapnia (Vermeire et al., 1987; Whipp, 1994). No hyperventilation was reported in response to sustained or progressive hypoxaemia, either at rest or during exercise (Holton & Wood, 1965). Ventilation did not change when subjects were exposed to hyperoxia (Lugliani et al., 1971). Abnormalities post-procedure persisted in the long term, and they tended to be more severe in cases of bilateral removal than unilateral removal (Honda, Watanabe et al., 1979).

Taken together, these findings suggest that while CBD may offer a therapeutic benefit in selecting pulmonary and cardiovascular conditions characterized by chemoreflex overactivity, its application must be approached with caution due to the essential role of the carotid body in maintaining respiratory homeostasis.

## Conclusions and perspectives

Chemoreflexes are potent respiratory and neural circulatory control modulators. It is yet unknown what causes aberrant chemoreflex function in pulmonary diseases. Dysfunction in systems that typically operate to modulate the chemoreflex under physiological settings is one of the possible explanations. These could be changes in the carotid bodies' chemosensory cells' sensitivity, unbalanced ROS, and redox stress, neuroplastic changes in the NTS, lamina terminalis, rostral ventrolateral medulla and PVN, elevated levels of circulating Ang II

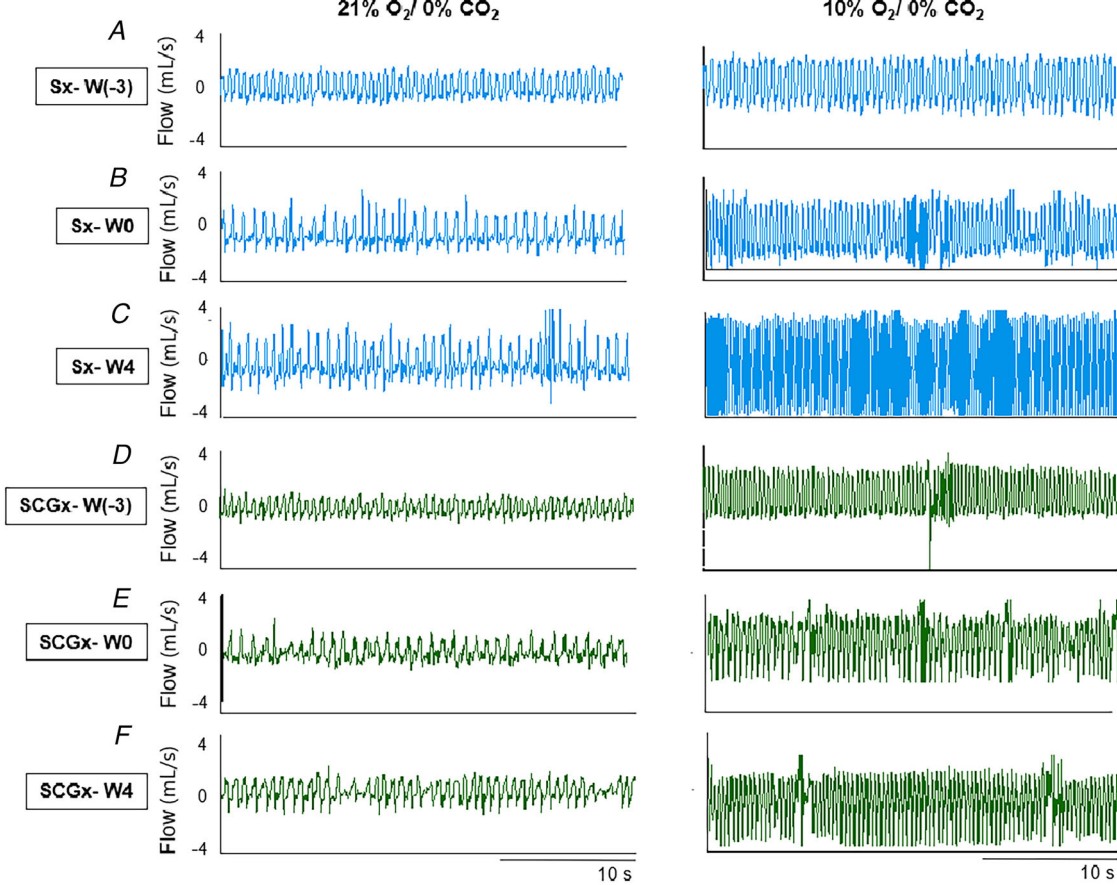

**Figure 4. Resting Breathing Under Normoxia and Hypoxia in Sx and SCGx Rats Over Time**
Representative recordings of resting breathing at normoxia (left) and 10% hypoxia (right) obtained in one rat per experimental group: (*A*), (*B*) and (*C*) Sx (in blue) at W0, W1 and W4, respectively; (*D*), (*E*) and (*F*) SCGx (in green) at W0, W1 and W4, respectively. Adapted from *Front Physiol. 2023 Feb 8;14:1 101 408* with permission.

and other pro-inflammatory cytokines, and/or neuro-inflammation. Future research should use transgenic reporter mice and techniques like tissue-clearing (Guo et al., 2022) that would convert intact carotid body tissues into hydrogel-tissue hybrids while keeping their anatomical structure, proteins and nucleic acids, better to understand the flexibility and mechanisms behind the diseases. *In vivo* calcium imaging and optogenetics (Pincus et al., 2021) could be used to examine ion channel functions and enzymes that may be involved in the remodelling or plasticity of neurons in various ganglia or brain regions. For instance, optogenetics, as well as DREADD (designer receptors exclusively activated by designer drugs) approaches, have been employed to study the chemoreflex as mediated by the neurons in the retrotrapezoid nucleus (SheikhBahaei et al., 2023) and rostral ventrolateral medulla (Abbott et al., 2013) neurons but a direct application of opto-/chemo-genetics in carotid body neurons remains unevaluated and could be explored in future studies. Moreover, an unbiased chromatin conformation capture (HI-C) approach in conjunction with DNA fluorescence *in situ* hybridization could be used to examine 3D genome organization (Rocks et al., 2022) in neurons at a deeper level. With more studies focusing on understanding 3D chromatin organization, it would be interesting to understand how the chromatin organization is changed between treatments by analysing compartments and loops at high resolution in carotid body tissues. The development of new technology now enables us to design experiments, study questions and find solutions to understand the mechanisms and develop advanced treatment strategies for these pulmonary diseases.

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

## Additional information

### Competing interests

None of the authors have any competing interests to disclose.

### Author contributions

K.K. drafted, revised the review article and prepared the figures; Z.X., I.H.Z., H.S. and H.J.W. provided constructive feedback, some new ideas, edited the text and approved the final version of the manuscript; All authors have read and approved the final version of the manuscript and all persons designated as authors qualify for authorship, and all those who qualify for authorship are listed.

### Funding

This study was supported by NIH grant R01 HL-152 160-01 and in part, by NIH grants R01 HL171602-01, HL-169 205-01, HL126796-01, HL172029-01A1 and R21 HL170127-01. H.J.W. is also supported by Margaret R. Larson Professorship in Anesthesiology. I.H.Z. is supported, in part, by the Theodore F. Hubbard Professorship for Cardiovascular Research.

### Keywords

acute lung injury, carotid bodies, chemoreflex, COPD, COVID-19, obstructive sleep apnoea

## Supporting information

Additional supporting information can be found online in the Supporting Information section at the end of the HTML view of the article. Supporting information files available:

**Peer Review History**

