## [Peer Review History · The Journal of Physiology]

Chemoreflex Function in Pulmonary Diseases- A Review

Kajal Kamra, Zhiqiu Xia, Irving H. Zucker, Harold D Schultz, and Han-Jun Wang

DOI: 10.1113/JP286655

Corresponding author(s): Han-Jun Wang (hanjunwang@unmc.edu)

The following individual(s) involved in review of this submission have agreed to reveal their identity: Nicholas Gregory Jendzjowsky (Referee #1); Silvia V Conde (Referee #2)

Review Timeline:

Submission Date:	27-Nov-2024
Editorial Decision:	14-Jan-2025
Revision Received:	06-May-2025
Editorial Decision:	03-Jun-2025
Revision Received:	26-Jun-2025
Accepted:	09-Jul-2025

Senior Editor: Laura Bennet

Reviewing Editor: Ken O'Halloran

Transaction Report:

Dear Dr Wang,

Re: JP-TR-2024-286655 "Chemoreflex Function in Pulmonary Diseases- A Review" by Kajal Kamra, Zhiqiu Xia, Irving H. Zucker, Harold D Schultz, and Han-Jun Wang

Thank you for submitting your manuscript to The Journal of Physiology. It has been assessed by a Reviewing Editor and by 2 expert referee(s) and we are pleased to tell you that it is potentially acceptable for publication following satisfactory major revision.

ABSTRACT FIGURES: Authors are expected to use The Journal's premium BioRender account to create/redraw their Abstract Figures. Information on how to access this account is here:

<https://physoc.onlinelibrary.wiley.com/journal/14697793/biorender-access>.

REVISION CHECKLIST:

IMPORTANT POINTS TO NOTE WHEN REVISING YOUR MANUSCRIPT:

LANGUAGE EDITING AND SUPPORT FOR PUBLICATION: If you would like help with English language editing, or other article preparation support, Wiley Editing Services offers expert help, including English Language Editing, as well as translation, manuscript formatting, and figure formatting at www.wileyauthors.com/eoo/preparation. You can also find resources for Preparing Your Article for general guidance about writing and preparing your manuscript at www.wileyauthors.com/eoo/prepresources.

We look forward to receiving your revised submission.

Yours sincerely,

Laura Bennet
Senior Editor
The Journal of Physiology

REQUIRED ITEMS

- Please include an Abstract Figure file, as well as the Figure Legend text within the main article file. The Abstract Figure is a piece of artwork designed to give readers an immediate understanding of the Review Article and should summarise the main conclusions. If possible, the image should be easily 'readable' from left to right or top to bottom. It should show the physiological relevance of the Review so readers can assess the importance and content of the article. Abstract Figures should not merely recapitulate other figures in the Review. Please try to keep the diagram as simple as possible and without superfluous information that may distract from the main conclusion of the Review. Abstract Figures must be provided by authors no later than the revised manuscript stage and should be uploaded as a separate file during online submission labelled as File Type 'Abstract Figure'. Please ensure that you include the figure legend in the main article file. All Abstract Figures will be sent to a professional illustrator for redrawing and you may be asked to approve the redrawn figure before your paper is accepted.

- Your MS must include a complete "Additional information section" with the following 4 headings and content:

Competing Interests: A statement regarding competing interests. If there are no competing interests, a statement to this effect must be included. All authors should disclose any conflict of interest in accordance with journal policy.

Author contributions: Each author should take responsibility for a particular section of the study and have contributed to writing the paper. Acquisition of funding, administrative support or the collection of data alone does not justify authorship; these contributions to the study should be listed in the Acknowledgements. Additional information such as 'X and Y have contributed equally to this work' may be added as a footnote on the title page.

It must be stated that all authors approved the final version of the manuscript and that all persons designated as authors qualify for authorship, and all those who qualify for authorship are listed.

Funding: Authors must indicate all sources of funding, including grant numbers. If authors have not received funding, this must be stated.

It is the responsibility of authors funded by RCUK to adhere to their policy regarding funding sources and underlying research material. The policy requires funding information to be included within the acknowledgement section of a paper. Guidance on how to acknowledge funding information is provided by the Research Information Network. The policy also requires all research papers, if applicable, to include a statement on how any underlying research materials, such as data, samples or models, can be accessed. However, the policy does not require that the data must be made open. If there are considered to be good or compelling reasons to protect access to the data, for example commercial confidentiality or legitimate sensitivities around data derived from potentially identifiable human participants, these should be included in the statement.

Acknowledgements: Acknowledgements should be the minimum consistent with courtesy. The wording of acknowledgements of scientific assistance or advice must have been seen and approved by the persons concerned. This section should not include details of funding.

- Please upload separate high quality figure files via the submission form.

- Author profile(s) must be uploaded via the submission form. Authors should submit a short biography (no more than 100 words for one author or 150 words in total for two authors) and a portrait photograph of the two leading authors on the paper. These should be uploaded and clearly labelled together in a Word document with the revised version of the manuscript. Any standard image format for the photograph is acceptable, but the resolution should be at least 300 DPI and

preferably more. A group photograph of all authors is also acceptable, providing the biography for the whole group does not exceed 150 words.

EDITOR COMMENTS

Reviewing Editor:

Thank you for submitting your topical review to The Journal of Physiology. Two experts in the field have reviewed your manuscript. Enthusiasm for the manuscript in its current form is mixed. In particular R2 raises concerns about some apparent misconceptions and also contends that the review at present is a collation of findings but needs revision to improve the flow of the narrative. Several important points are raised by the referees but I am confident that these can be addressed. I encourage you to carefully consider and address the referee's critiques and to revise the manuscript in accordance with the concerns raised and with a view to improving the clarity of the message.

Please also see 'Required Items' above.

REFEREE COMMENTS

Referee #1:

The review by Kamra et al. provides information on chemoreflex (dys)function in respiratory diseases. There is a substantial amount of information and consolidation into an updated review is warranted. The authors provide a good effort to consolidate a vast amount of information in a limited space. The figures assist the reader with this understanding. Overall this is good manuscript.

There are a number of comments which need to be addressed:

MAJOR:

Given the nature of ALI, COPD and asthma being diseases/syndromes with aberrant immune responses, carotid body sensitivity to cytokines (PMID: 35339628,33180962) and inflammatory lipids (PMID: 30279412) is worthy of discussion. The authors provide discussion of DAMPS but not PAMPS.

The authors touch on how inflammation may alter chemoreflex drive in the conclusion. This could be expanded.

MINOR:

Line 45- should read when blood, not when the body.

Line 48- CIH is not a pathological condition. Sleep apnea, due to CIH is.

Line 50- mention of high altitude may not be relevant to this review and this comment disrupts the flow of the paragraph.

Line 54- reference for CO₂ responsiveness of brainstem required. New evidence also show O₂ sensitivity, see reviews from Funk and Gourine.

Line 59- references for each cardiovascular, renal, endocrine and immune function should be included.

Line68-69- The statement that less research into the role of chemoreflex function in cardiorespiratory disease is not necessarily true. There has been a vast amount of research from the 1960s-1980s on asthma and COPD in regards to carotid body activity. Nadel and Widdicombe have extensively researched this.

Line 86- Humans lack TH in carotid bodies? reference?

Line 93-96 is confusing. As it reads, this reviewers interpretation is that type A cells along with Type A cells forms a

synapse? Clarification is required.

Line 105-106 is confusing. The premise of the review by Zera et al. is that either clusters of glomus cells respond to certain stimuli.

Line 124- reference required.

Line 156-161- To the reviewer's knowledge there is only some evidence to show that there is a depressed ventilatory response to hypoxia in COVID patients. Others (Malhotra) have provided evidence to refute this. That virus was found (in a case study) in human carotid body tissue does not provide evidence of carotid chemoreflex malfunction. Unless HVR/HCVR was measured in COVID patients, substantial evidence of chemoreflex perturbations is not available.

Line 181- there are a number of studies showing carotid body sensitivity to cytokines where some suppress and some increase carotid body excitation. Please reference.

Line 273- actually reference fletcher et al, del rio et al. . .

Line 320- NO stimulating ATP release needs a reference

Line 320-321- mentions Bock et al. but references Somers?

Referee #2:

The manuscript by Kamra et al. seeks to review the role of chemoreflexes, particularly the carotid bodies, in pulmonary diseases. While it touches on potentially important areas such as COVID-19, acute lung injury, and COPD, it also revisits the widely reviewed topic of chronic intermittent hypoxia and obstructive sleep apnea.

However, the manuscript lacks the necessary quality, clarity, and coherence. It is more of a compilation of information than a cohesive effort to propose explanatory theories or offer novel insights. Moreover, it contains incorrect concepts, numerous improperly cited references, and outdated citations in sections where more current references would significantly enhance its relevance and accuracy.

Please find below my criticisms to the manuscript and suggestions for improvement:

1. Page 3 line 40-42 referring to Heymans Nobel prize - Please rephrase as Heymans won his Nobel Prize for the discovery of the role played by the sinus and aortic mechanisms in the regulation of respiration.
2. Page 4 line 60 - The concept of the carotid sinus nerve is introduced without explanation. Please define it
3. Page 4 line 61 to 63 - Link this sentence with the increased carotid sinus nerve activity
4. The section anatomy and physiology of chemoreflex during physiological conditions has multiple miss concepts.
 - 4.1. The consensual terminology for carotid body cells is type 1 and 2 cells, or glomus and sustentacular cells but never type A and B. If the authors are talking only about the type 1 cells and its different subpopulations (based on dense core vesicles - that have been made in rats) this must be better described throughout the manuscript and not be misunderstood with the role of the neurogenic niche for example in chronic hypoxic conditions.
 - 4.2. In line 80 the authors speak about the plasticity of type 2 cells under sustained hypoxia but note that alterations in the neurogenic niche have been also documented in CIH (Caballero-Eraso 2023 J physiol).

4.3. In this section please also include the reference for the decreased expression of TH in humans (Ortega-Sanz 2013 J physiol <https://doi.org/10.1113/jphysiol.2013.263657>).

4.4. Moreover, a more updated information on the communication between the different types of cells should be done - the tripartite synapse, see e.g. Leonard et al. 2018 DOI: 10.3389/fphys.2018.00225; Leonard and Nurse, 2023 doi: 10.1007/978-3-031-32371-3_20.

4.5. Lines 101 to 104 - The authors assume that different clusters of type 1 cells have different sensitivities to hypoxia and pH based on the manuscript by Zera et al. 2019. However, that statement is based on the data from Lu et al. (2013 J physiol DOI: 10.1113/jphysiol.2012.247189) where the authors show in mice there are some clusters that exhibit sensitivity only to hypoxia and others to low pH, but that the majority of the clusters show sensitivity to both. It remains uncertain if this happens for all the stimuli sensed by the carotid body and if for all the mammal species.

4.6. Line 107 and 108 on the type of neurotransmitters present in carotid body type 1 cells - The discussion on carotid body neurotransmitters should reference broader literature beyond Zera et al., 2019 E.g. <https://doi.org/10.1152/physrev.00039.2019>

4.7. Lines 122 - 125, please state to which sites outside the NTS the CB afferents might project.

5. Section Chemoreflex in sars-cov 2

5.1. Lines 156-158 - it is not clear for this review why the manuscript of Iring et al. 2024 entitled "Blood oxygen regulation via P2Y12R expressed in the carotid body" is cited in this sentence

5.2. Lines 181- 182 - there is an extensive literature on carotid body and inflammation. It is consensual that the CB expresses receptors for pro and anti-inflammatory cytokines in animals (mice, rat) and humans, that proinflammatory cytokines modulate the activity and function of carotid body cells. See for example: Iturriaga et al. 2021 doi: 10.1152/physiol.00031.2021; Iturriaga 2023 doi: 10.1113/JP284112, Sacramento et al. 2020, DOI: 10.3390/ijms21155545). Therefore, this topic must be more explored.

6. Section Chronic intermittent hypoxia and obstructive sleep apnea

6.1. - References on lines 224-225 - Cite Peng et al. (2003, PNAS, DOI: 10.1073/pnas.1734109100) for the first description of long-term facilitation of carotid body activity.

6.2. Lines 258-261 - please make it more comprehensive

6.3. Lines 272-275 statement " In a detailed review by Kara et al, the author summarizes studies by Fletcher et al., Peng et al., and Del Rio et al., who found strong evidence of the involvement of sensitized carotid body chemoreflex in CIH-exposed hypertension and/or OSA rodent and cat models (Kara et al., 2003)." Note that only the manuscripts by Fletcher are previous to the manuscript by Kara in 2003. Therefore, it is impossible that the review by Kara included the studies by Peng and Del Rio.

6.4. Please change hif-1a and Hif1-b to HIF-1 alpha and HIF-1beta.

7. The work by Jendzjowsky et al. 2018 Nat Communications (10.1038/s41467-018-06189-y) that provides a link between carotid body-mediated bronchoconstriction and allergen-induced asthma as well as the work by the same authors (2021 J Neuroinflammation DOI: 10.1186/s12974-021-02241-9) showing the ability of allergens to sensitize the carotid bodies, highlighting the likely role of the carotid bodies and blood-borne inflammatory mediators in asthma should be considered.

8. Section on carotid sinus nerve denervation - there are multiple studies and more recent focused on the adverse effects of carotid sinus nerve that deserve to be cited.

Minor points

There are multiple typos throughout the manuscript. E.g. abstract - page 2 line 26 - it should be "the current review summarizes" and not "the current review will summarize".

END OF COMMENTS

Response to Reviewers

Dear Reviewers,

Thanks very much for your excellent comments. Your comments will greatly improve the quality of this manuscript. We have made a revision according to these comments. The details are described in a point-to-point way:

Referee #1:

The review by Kamra et al. provides information on chemoreflex (dys)function in respiratory diseases. There is a substantial amount of information and consolidation into an updated review is warranted. The authors provide a good effort to consolidate a vast amount of information in a limited space. The figures assist the reader with this understanding. Overall this is good manuscript.

There are a number of comments which need to be addressed:

MAJOR:

1. Given the nature of ALI, COPD and asthma being diseases/syndromes with aberrant immune responses, carotid body sensitivity to cytokines (PMID: 35339628,33180962) and inflammatory lipids (PMID: 30279412) is worthy of discussion. The authors provide discussion of DAMPS but not PAMPS.

2. The authors touch on how inflammation may alter chemoreflex drive in the conclusion. This could be expanded.

Response for 1. and 2.: To accommodate both Reviewer 1 and Reviewer 2's comments about enhancing discussion on the carotid body and inflammation, we have added a new subsection, "Carotid body sensitivity to inflammatory mediators" that discusses this topic in some detail. Please see L193-221.

MINOR:

3. Line 45- should read when blood, not when the body.

Response: Thank you for your comment. This change has been incorporated into the manuscript. Please see L45.

4. Line 48- CIH is not a pathological condition. Sleep apnea, due to CIH is.

Response: Thank you for your comment. This change has been incorporated into the manuscript. Please see L48-49.

5. Line 50- mention of high altitude may not be relevant to this review, and this comment disrupts the flow of the paragraph.

Response: Thank you for your comment. We agree and have deleted this line from the manuscript.

6. Line 54- reference for CO₂ responsiveness of brainstem required. New evidence also shows O₂ sensitivity, see reviews from Funk and Gourine.

Response: Thank you for your comment. We have added this to the manuscript. Please see L51-58.

7. Line 59- references for each cardiovascular, renal, endocrine, and immune function should be included.

Response: All references suggested have been added into the manuscript. Please see L63-64.

8. Line68-69- The statement that less research into the role of chemoreflex function in cardiorespiratory disease is not necessarily true. There has been a vast amount of research from the 1960s-1980s on asthma and COPD in regards to carotid body activity. Nadel and Widdicombe have extensively researched this.

Response: We appreciate you pointing this out. We have made appropriate changes in this regard. Please see L74-82.

9. Line 86- Humans lack TH in carotid bodies? reference?

Response: We have added a reference at the end of the original sentence in the manuscript. Please see L109.

10. Line 93-96 is confusing. As it reads, this reviewers interpretation is that type A cells along with Type A cells forms a synapse? Clarification is required.

Response: Thank you. This section has been rewritten to avoid confusion. Please see L92-131

11. Line 105-106 is confusing. The premise of the review by Zera et al. is that either clusters of glomus cells respond to certain stimuli.

Response: This section has been rewritten to avoid confusion. Please see L92-131

12. Line 124- reference required.

Response: This line has been modified in the revised version of the manuscript, and a reference has been added.

13. Line 156-161- To the reviewer's knowledge there is only some evidence to show that there is a depressed ventilatory response to hypoxia in COVID patients. Others (Malhotra) have provided evidence to refute this. That virus was found (in a case study) in human carotid body tissue does not provide evidence of carotid chemoreflex malfunction. Unless HVR/HCVR was measured in COVID patients, substantial evidence of chemoreflex perturbations is not available.

Response: We agree with you! This information has been added to the manuscript. Please see L232-235.

14. Line 181- there are a number of studies showing carotid body sensitivity to cytokines where some suppress and some increase carotid body excitation. Please reference.

Response: This paragraph has been modified to accommodate both reviewers 1 and 2's comments. Please see L193-221

15. Line 273- actually reference fletcher et al, del rio et al. . .

Response: Thank you. We have added the correct references to the manuscript. Please see L346-348

16. Line 320- NO stimulating ATP release needs a reference

Response: Thanks for pointing that out. Upon searching for the citation, we realized that this claim is not entirely correct. So, we have rephrased this sentence to avoid confusion. Please see L395-398.

17. Line 320-321- mentions Bock et al. but references Somers?

Response: Thanks for catching that. The wrong citation has been replaced with the correct one. Please see L398.

Referee #2:

The manuscript by Kamra et al. seeks to review the role of chemoreflexes, particularly the carotid bodies, in pulmonary diseases. While it touches on potentially important areas such as COVID-19, acute lung injury, and COPD, it also revisits the widely reviewed topic of chronic intermittent hypoxia and obstructive sleep apnea.

However, the manuscript lacks the necessary quality, clarity, and coherence. It is more of a compilation of information than a cohesive effort to propose explanatory theories or offer novel insights. Moreover, it contains incorrect concepts, numerous improperly cited references, and outdated citations in sections where more current references would significantly enhance its relevance and accuracy.

Please find below my criticisms to the manuscript and suggestions for improvement:

1. Page 3 line 40-42 referring to Heymans Nobel prize - Please rephrase as Heymans won his Nobel Prize for the discovery of the role played by the sinus and aortic mechanisms in the regulation of respiration.

Response: Thank you for your comment. This change has been incorporated into the manuscript. Please see L41-42.

2. Page 4 line 60 - The concept of the carotid sinus nerve is introduced without explanation. Please define it

Response: This change has now been incorporated into the manuscript. Please see L63-71

3. Page 4 line 61 to 63 - Link this sentence with the increased carotid sinus nerve activity

Response: We have rewritten this sentence in the manuscript. Please see L63-71

4. The section anatomy and physiology of chemoreflex during physiological conditions has multiple miss concepts.

4.1. The consensual terminology for carotid body cells is type 1 and 2 cells, or glomus and sustentacular cells but never type A and B. If the authors are talking only about the type 1 cells and its different subpopulations (based on dense core vesicles - that have been made in rats) this must be better described throughout the manuscript and not be misunderstood with the role of the neurogenic niche for example in chronic hypoxic conditions.

Response: To accommodate both Reviewer 1 and Reviewer 2's comments about enhancing discussion on the carotid body and inflammation, I have added a new subsection that discusses this topic in some detail. Please see L193-221.

4.2. In line 80 the authors speak about the plasticity of type 2 cells under sustained hypoxia but note that alterations in the neurogenic niche have been also documented in CIH (Caballero-Eraso 2023 J physiol).

Response: This change has been incorporated into the manuscript. Please see L99.

4.3. In this section, please also include the reference for the decreased expression of TH in humans (Ortega-Sanz 2013 J physiol <https://doi.org/10.1113/jphysiol.2013.263657> [doi.org]).

Response: Thank you for your comment. We have added a reference at the end of the original sentence in the manuscript. Please see L109.

4.4. Moreover, more updated information on the communication between the different types of cells should be done - the tripartite synapse, see e.g. Leonard et al. 2018 DOI: 10.3389/fphys.2018.00225; Leonard and Nurse, 2023 doi: 10.1007/978-3-031-32371-3_20.

Response: As suggested we have included a discussion about the tripartite synapse. Please L114-125.

4.5. Lines 101 to 104 - The authors assume that different clusters of type 1 cells have different sensitivities to hypoxia and pH based on the manuscript by Zera et al. 2019. However, that statement is based on the data from Lu et al. (2013 J physiol DOI: 10.1113/jphysiol.2012.247189) where the authors show in mice there are some clusters that exhibit sensitivity only to hypoxia and others to low pH, but that the majority of the clusters show sensitivity to both. It remains uncertain if this happens for all the stimuli sensed by the carotid body and if for all the mammal species.

Response: Thanks for pointing that out. We have made a reference of that in the manuscript. Please see L143-145.

4.6. Line 107 and 108 on the type of neurotransmitters present in carotid body type 1 cells - The discussion on carotid body neurotransmitters should reference broader literature beyond Zera et al., 2019 E.g. <https://doi.org/10.1152/physrev.00039.2019> [doi.org]

Response: As suggested we have added additional references to support this sentence in the manuscript. Please see L146.

4.7. Lines 122 - 125, please state to which sites outside the NTS the CB afferents might project.

Response: We have added a reference at the end of the original sentence in the manuscript. Please see L143-144.

5. Section Chemoreflex in sars-cov 2

5.1. Lines 156-158 - it is not clear for this review why the manuscript of Iring et al. 2024 entitled "Blood oxygen regulation via P2Y12R expressed in the carotid body" is cited in this sentence

Response: Thanks for pointing that out. The incorrect citation has been removed from the manuscript. Please see L45.

5.2. Lines 181- 182 - there is an extensive literature on carotid body and inflammation. It

is consensual that the CB expresses receptors for pro and anti-inflammatory cytokines in animals (mice, rat) and humans, that proinflammatory cytokines modulate the activity and function of carotid body cells. See for example: Iturriaga et al. 2021 doi: 10.1152/physiol.00031.2021; Iturriaga 2023 doi: 10.1113/JP284112, Sacramento et al. 2020, DOI: 10.3390/ijms21155545). Therefore, this topic must be more explored.

Response: To accommodate both Reviewer 1 and Reviewer 2's comments about enhancing discussion on the carotid body and inflammation, we have added a new subsection, "Carotid body sensitivity to inflammatory mediators" that discusses this topic in some detail. Please see L193-221.

6. Section Chronic intermittent hypoxia and obstructive sleep apnea

6.1. - References on lines 224-225 - Cite Peng et al. (2003, PNAS, DOI: 10.1073/pnas.1734109100) for the first description of long-term facilitation of carotid body activity.

Response: We have added a reference at the end of the original sentence in the revised manuscript. Please see L296.

6.2. Lines 258-261 - please make it more comprehensive

Response: We have rephrased these lines to make them more comprehensive. Please see L328-336.

6.3. Lines 272-275 statement " In a detailed review by Kara et al, the author summarizes studies by Fletcher et al., Peng et al., and Del Rio et al., who found strong evidence of the involvement of sensitized carotid body chemoreflex in CIH-exposed hypertension and/or OSA rodent and cat models (Kara et al., 2003)." Note that only the manuscripts by Fletcher are previous to the manuscript by Kara in 2003. Therefore, it is impossible that the review by Kara included the studies by Peng and Del Rio.

Response: Thank you for catching this. We have made the appropriate changes to this section. Please see L346-351.

6.4. Please change hif-1a and Hif1-b to HIF-1 alpha and HIF-1beta.

Response: Changed. Please see L375.

7. The work by Jendzjowsky et al. 2018 Nat Communications (10.1038/s41467-018-06189-y) that provides a link between carotid body-mediated bronchoconstriction and allergen-induced asthma as well as the work by the same authors (2021 J Neuroinflammation DOI: 10.1186/s12974-021-02241-9) showing the ability of allergens to sensitize the carotid bodies, highlighting the likely role of the carotid bodies and blood-borne inflammatory mediators in asthma should be considered.

Response: We have included these papers in our discussion. Please see L522-534.

8. Section on carotid sinus nerve denervation - there are multiple studies and more recent focused on the adverse effects of carotid sinus nerve that deserve to be cited.

Response: Thanks for your suggestion. We have rewritten this subsection to accommodate your comment. Please see L583-630.

Minor points

9. There are multiple typos throughout the manuscript. E.g. abstract - page 2 line 26 - it should be "the current review summarizes" and not "the current review will summarize".

Response: Thanks for your comment. We have made this change to the manuscript and proofread the latest draft for any other typos.

Dear Dr Wang,

Re: JP-TR-2025-286655R1 "Chemoreflex Function in Pulmonary Diseases- A Review" by Kajal Kamra, Zhiqiu Xia, Irving H. Zucker, Harold D Schultz, and Han-Jun Wang

Thank you for submitting your manuscript to The Journal of Physiology. It has been assessed by a Reviewing Editor and by 2 expert referees and we are pleased to tell you that it is acceptable for publication following satisfactory revision.

ABSTRACT FIGURES: Authors may use The Journal's premium BioRender account to create/redraw their Abstract Figures (and any other suitable schematic figure). Information on how to access this account is here: <https://physoc.onlinelibrary.wiley.com/journal/14697793/biorender-access>.

REVISION CHECKLIST: Upload a full Response to Referees file. To create your 'Response to Referees' copy all the reports, including any comments from the Senior and Reviewing Editors, into a Microsoft Word, or similar, file and respond to each point, using font or background colour to distinguish comments and responses and upload as the required file type.

We look forward to receiving your revised submission.

Yours sincerely,

Laura Bennet
Senior Editor

EDITOR COMMENTS

Reviewing Editor:

Thank you for making revisions to the original text. R1 is satisfied with the manuscript in its current form, but R2 has additional valid comments and suggestions. I invite you to address these final outstanding concerns and agree that some additional editing should help to improve the flow of the narrative. The review is timely and likely to be an important resource for researchers in the field. We look forward to receiving a revised version of the manuscript.

REFEREE COMMENTS

Referee #1:

The authors have adequately addressed this reviewer's concerns.

Referee #2:

The authors have addressed some of the changes requested by this reviewer however, I still believe that the manuscript still has room for improvement. Note that some sections lack fluidity and some of the ideas that were suggested to be included are not in the correct place within the manuscript.

Please see my specific comments below:

- Line 66, page 4 - from what is written it seems that the carotid sinus nerve only enervates the carotid body "in the case of a sensitized peripheral chemoreflex". Please rephrase
- Lines 80-84 are a repetition of what is written between lines 72-79, but with less detail.
- The section Anatomy and physiology of chemoreflex during physiological conditions could be better written and integrated in the manuscript. From lines 114 to 120, the authors talk about the tripartite synapse and the role of ATP on this synapse without introducing ATP as a neurotransmitter within the carotid body. Maybe the section between 125-144 can come before lines 106-124.
- When talking about the different sensitivity of type 1 cell clusters to hypoxia and low pH please cite the new manuscript by Spiller et al. Short-term sustained hypoxia distinctly affects subpopulations of carotid body glomus cells from rats. *Am J Physiol Cell Physiol.* 2025 doi: 10.1152/ajpcell.00967.2024.
- Line 149 authors state glutamate as a carotid body transmitter, please include references.
- Line 164 please rephrase as Zera is not from Paton group.
- Line 196-187 please detail more the effect of different cytokines on carotid body function as not all have the same effects in K⁺ currents, intracellular calcium and carotid sinus nerve activity.
- Section dedicated to Carotid Body Sensitivity to Inflammatory Mediators, the authors should include the reference from the seminal studies where the effects of cytokines in the carotid body were described.
- From my point of view, the section Chronic Intermittent Hypoxia and Obstructive Sleep Apnea should start by describing

the OSA association with altered chemoreflexes and then describe the effects of CIH, as CIH is a component of OSA.

- Line 313 which animal models are the authors talking about? A CIH model?

- Line 316, please describe the situation were people are exposed to CIH to increase clarity

- Line 423-424 note that the manuscript by Olea et al. 2024 do not shows that COPD increases sympathetic activity as it deals study pulmonary vascular responses to chronic intermittent hypoxia in a guinea pig model of obstructive sleep apnea.

- Authors should consider to increase the number of figures in the manuscript to increase clarity

END OF COMMENTS

Response to Reviewer #2: Thank you for your additional comments. We have addressed each point individually as outlined below.

The authors have addressed some of the changes requested by this reviewer however, I still believe that the manuscript still has room for improvement. Note that some sections lack fluidity and some of the ideas that were suggested to be included are not in the correct place within the manuscript.

Please see my specific comments below:

Comment 1- Line 66, page 4 - from what is written it seems that the carotid sinus nerve only innervates the carotid body "in the case of a sensitized peripheral chemoreflex". Please rephrase

Response: Thank you for your comment. This sentence has been rephrased in the revised manuscript. Please see L63-67.

Comment 2- Lines 80-84 are a repetition of what is written between lines 72-79, but with less detail.

Response: Thank you for your comment. This sentence has been removed in the revised manuscript.

Comment 3- The section Anatomy and physiology of chemoreflex during physiological conditions could be better written and integrated in the manuscript. From lines 114 to 120, the authors talk about the tripartite synapse and the role of ATP on this synapse without introducing ATP as a neurotransmitter within the carotid body. Maybe the section between 125-144 can come before lines 106-124.

Response: Thank you for your comment. This subsection has been reformatted. Please note that L101-129 has been moved before L130 in the revised manuscript.

Comment 4- When talking about the different sensitivity of type 1 cell clusters to hypoxia and low pH please cite the new manuscript by Spiller et al. Short-term sustained hypoxia distinctly affects subpopulations of carotid body glomus cells from rats. Am J Physiol Cell Physiol. 2025 doi: 10.1152/ajpcell.00967.2024.

Response: Thank you for your comment. We have added this reference in the revised manuscript. Please see L112.

Comment 5- Line 149 authors state glutamate as a carotid body transmitter, please include references.

Response: Thank you for your comment. We have added a reference to support this sentence in the revised manuscript. Please see L125.

Comment 6- Line 164 please rephrase as Zera is not from Paton group.

Response: Thank you for your comment. We have made this change in the revised manuscript. Please see L158-159.

Comment 7- Line 196-187 please detail more the effect of different cytokines on carotid body function as not all have the same effects in K⁺ currents, intracellular calcium and carotid sinus nerve activity.

Response: Thank you for your comment. We have revised this subsection in the revised manuscript. Please see L191-205.

Comment 8- Section dedicated to Carotid Body Sensitivity to Inflammatory Mediators, the authors should include the reference from the seminal studies where the effects of cytokines in the carotid body were described.

Response: Thank you for your comment. We have added the appropriate references in the revised manuscript. Please see L189-221

Comment 9- From my point of view, the section Chronic Intermittent Hypoxia and Obstructive Sleep Apnea should start by describing the OSA association with altered chemoreflexes and then describe the effects of CIH, as CIH is a component of OSA.

Response: Thank you for your comment. We have changed the subheading for this subsection to “Obstructive Sleep Apnea and Chronic Intermittent Hypoxia” and have moved the text around for a better flow. Please see L283-322 in the revised manuscript.

Comment 9- Line 313 which animal models are the authors talking about? A CIH model?

Response: Thank you for your comment. The studies mentioned talk about CIH models. We have rephrased that sentence to add more clarity and have also added citations for the studies mentioned in the following lines. Please see L335, 358, and 360 in the revised manuscript.

Comment 10- Line 316, please describe the situation where people are exposed to CIH to increase clarity

Response: Thank you for your comment. We have made this change to the revised manuscript. Please see L319-322.

Comment 11- Line 423-424 note that the manuscript by Olea et al. 2024 does not show that COPD increases sympathetic activity as it deals with study pulmonary vascular responses to chronic intermittent hypoxia in a guinea pig model of obstructive sleep apnea.

Response: Thank you for your comment. We have corrected this in the revised manuscript. Please see L418.

Comment 12- Authors should consider to increase the number of figures in the manuscript to increase clarity

Response: Thank you for your comment. We appreciate your suggestion and understand the importance of visual elements in enhancing reader comprehension. In our current submission, we have dedicated one of the two figures as an abstract figure to visually summarize the key conceptual framework of the review, while the second figure comprehensively integrates the main mechanistic themes discussed throughout the manuscript. We have added three additional figures to support our Chronic intermittent hypoxia and acute lung injury subsections of the manuscript. Please see Figures 2, 3, and 4 on L347, 553, and 572, respectively.

Dear Dr Wang,

Re: JP-TR-2025-286655R2 "Chemoreflex Function in Pulmonary Diseases- A Review" by Kajal Kamra, Zhiqiu Xia, Irving H. Zucker, Harold D Schultz, and Han-Jun Wang

We are pleased to tell you that your paper has been accepted for publication in The Journal of Physiology.

Authors should note that it is too late at this point to offer corrections prior to proofing. Major corrections at proof stage, such as changes to figures, will be referred to the Editors for approval before they can be incorporated. Only minor changes, such as to style and consistency, should be made at proof stage. Changes that need to be made after proof stage will usually require a formal correction notice.

Yours sincerely,

Laura Bennet
Senior Editor
The Journal of Physiology

P.S. - You can help your research get the attention it deserves! Check out Wiley's free Promotion Guide for best-practice recommendations for promoting your work at www.wileyauthors.com/eoo/guide. You can learn more about Wiley Editing Services which offers professional video, design, and writing services to create shareable video abstracts, infographics, conference posters, lay summaries, and research news stories for your research at www.wileyauthors.com/eoo/promotion.

IMPORTANT NOTICE ABOUT OPEN ACCESS: To assist authors whose funding agencies mandate public access to published research findings sooner than 12 months after publication, The Journal of Physiology allows authors to pay an Open Access (OA) fee to have their papers made freely available immediately on publication.

You can check if your funder or institution has a Wiley Open Access Account here: <https://authorservices.wiley.com/author-resources/Journal-Authors/licensing-and-open-access/open-access/author-compliance-tool.html>.

EDITOR COMMENTS

Reviewing Editor:

Thank you for making these final changes to the manuscript. All issues have now been addressed. This timely and informative review will likely prove important for the field and should be of significant interest to the readership of The Journal of Physiology. Thank you for this insightful contribution.

REFeree COMMENTS

Referee #2:

The authors have addressed my previous comments and added figures that improve the clarity of the manuscript. I have no further major concerns, but I recommend carefully reviewing the use of abbreviations, as some are introduced only after their first appearance in the text.